# A novel nematode species from the Siberian permafrost shares adaptive mechanisms for cryptobiotic survival with *C. elegans* dauer larva

Anastasia Shatilovich[1,2°], Vamshidhar R. Gade[3,4°], Martin Pippel[5], Tarja T. Hoffmeyer[6], Alexei V. Tchesunov[7], Lewis Stevens[8], Sylke Winkler[3,9], Graham M. Hughes[10], Sofia Traikov[3], Michael Hiller[5,11], Elizaveta Rivkina[1], Philipp H. Schiffer[6]*, Eugene W. Myers[5], Teymuras V. Kurzchalia[3]*

1 Institute of Physicochemical and Biological Problems in Soil Science RAS, Pushchino, Russia, 2 Zoological Institute RAS, St. Petersburg, Russia, 3 Max Planck Institute for Molecular Cell Biology and Genetics, Dresden, Germany, 4 Institute of Biochemistry, ETH Zürich, Zürich, Switzerland, 5 Center for Systems Biology, Dresden, Germany, 6 Institute for Zoology, University of Cologne, Köln, Germany, 7 Department of Invertebrate Zoology, Lomonosov Moscow State University, Moscow, Russia, 8 Tree of Life, Wellcome Sanger Institute, Cambridge, United Kingdom, 9 DRESDEN concept Genome Center, Dresden, Germany, 10 School of Biology and Environmental Science, University College Dublin, Belfield, Dublin, Ireland, 11 LOEWE Centre for Translational Biodiversity Genomics, Senckenberg Society for Nature Research & Goethe University, Frankfurt am Main, Germany

° These authors contributed equally to this work.
* p.schiffer@uni-koeln.de (PHS); kurzchalia@mpi-cbg.de (TVK)

**Data Availability Statement:** All the data files are available at https://doi.org/10.5281/zenodo.6590382.

## Abstract

Some organisms in nature have developed the ability to enter a state of suspended metabolism called cryptobiosis when environmental conditions are unfavorable. This state-transition requires execution of a combination of genetic and biochemical pathways that enable the organism to survive for prolonged periods. Recently, nematode individuals have been reanimated from Siberian permafrost after remaining in cryptobiosis. Preliminary analysis indicates that these nematodes belong to the genera *Panagrolaimus* and *Plectus*. Here, we present precise radiocarbon dating indicating that the *Panagrolaimus* individuals have remained in cryptobiosis since the late Pleistocene (~46,000 years). Phylogenetic inference based on our genome assembly and a detailed morphological analysis demonstrate that they belong to an undescribed species, which we named *Panagrolaimus kolymaensis*. Comparative genome analysis revealed that the molecular toolkit for cryptobiosis in *P. kolymaensis* and in *C. elegans* is partly orthologous. We show that biochemical mechanisms employed by these two species to survive desiccation and freezing under laboratory conditions are similar. Our experimental evidence also reveals that *C. elegans* dauer larvae can remain viable for longer periods in suspended animation than previously reported. Altogether, our findings demonstrate that nematodes evolved mechanisms potentially allowing them to suspend life over geological time scales.

**Funding:** This work was supported by the Russian Foundation for Basic Research (19-29-05003-mk) to AS and ER. VRG and TVK acknowledge the financial support from the Volkswagen Foundation (Life research grant 92847). PHS and TTH are supported by a DFG ENP grant to PHS (DFG project 434028868). GMH is funded by a UCD Ad Astra Fellowship. The funders had no role in study design, data collection and analysis, decision to publish, or preparation of the manuscript.

**Competing interests:** The authors have declared that no competing interests exist.

## Author summary

Survival in extreme environments for prolonged periods is a challenge that only a few organisms, are capable of. It is not well understood, which molecular and biochemical pathways are utilized by such cryptobiotic organisms, and how long they might suspend life. Here, we show that a soil nematode *Panagrolaimus kolymaensis*, suspended life for 46,000 years in the Siberian permafrost. Through comparative analysis, we find that *P. kolymaensis* and model organism *C. elegans* utilize similar adaptive mechanisms to survive harsh environmental conditions for prolonged periods. Our findings here are important for the understanding of evolutionary processes because generation times could be stretched from days to millennia, and long-term survival of individuals of species can lead to the refoundation of otherwise extinct lineages.

## Introduction

Organisms from diverse taxonomic groups can survive extreme environmental conditions, such as the complete absence of water or oxygen, high temperature, freezing, or extreme salinity. The survival strategies of such organisms include a state known as suspended animation or cryptobiosis, in which they reduce metabolism to an undetectable level [1]. Spectacular examples of long-term cryptobiosis include a *Bacillus* spore that was preserved in the abdomen of bees buried in amber for 25 to 40 million years [2], and a 1000 to 1500 years-old *Lotus* seed, found in an ancient lake, that was subsequently able to germinate [3]. Metazoans such as tardigrades, rotifers, and nematodes are also known for remaining in cryptobiosis for prolonged periods [4,5]. The longest records of cryptobiosis in nematodes are reported for the Antarctic species *Plectus murrayi* [6] (25.5 years in moss frozen at -20˚C), and *Tylenchus polyhypnus* [7] (39 years desiccated in an herbarium specimen).

Intensive research during the last decade has demonstrated that permafrostg (perennially frozen sediments) are unique ecosystems preserving life forms at sub-zero temperatures over thousands of years [8,9,10,11]. Permafrost remains are an exceptional source for discovering a wide variety of unicellular and multicellular living organisms surviving in cryptobiosis for prolonged periods [1,12,13]. The Siberian permafrost is a unique repository for preserving organisms in sub-zero temperatures for millions of years. Expeditions in the past decade have resulted in the revival of several organisms across various taxa from the Siberian permafrost [14,15,16,17]. The possibility to exploit permafrost as a source for reanimating multicellular animals was recognized as early as 1936. A viable Cladocera crustacean, *Chydorus sphaericus*, preserved in the Transbaikalian permafrost for several thousand years [18,19], was discovered by P. N. Kapterev, who worked at the scientific station Skovorodino as a GULAG prisoner. Unfortunately, this observation remained unnoticed for many decades. We recently reanimated soil nematodes that were preserved in Siberian permafrost for potentially thousands of years, and initial morphological observations provisionally described them as belonging to the genera *Panagrolaimus* and *Plectus*. Previous studies demonstrated several species of *Panagrolaimus* can undergo cryptobiosis in the form of anhydrobiosis (through desiccation) and cryobiosis (through freezing) [20,21,22,23,24]. In various nematodes, entry into anhydrobiosis is often accompanied by a preparatory phase of exposure to mild desiccation, known as preconditioning [22,25]. This induces a specific re-modelling of the transcriptome, the proteome, and metabolic pathways that enhances survival ability [26,27,28]. Some panagrolaimids possess adaptive mechanisms for rapid desiccation where most of the cellular water is lost, while others

possess freezing tolerance without loss of water at sub-zero temperatures by inhibiting the growth and recrystallisation of ice crystals [22].

Here, we present a high-quality genome assembly, detailed morphological phylogenetic analysis, and define a novel species, *Panagrolaimus kolymaensis*. Precise radiocarbon dating indicates that *P. kolymaensis* remained in cryptobiosis for about 46,000 years, since the late Pleistocene. Making use of the model organism *C. elegans*, we demonstrate that *C. elegans* dauer larvae and *Panagrolaimus kolymaensis* utilize comparable molecular mechanisms to survive extreme desiccation and freezing, i.e. upregulation of trehalose biosynthesis and gluconeogenesis.

## Results

### Taxonomy

**Panagrolaimus kolymaensis** Shatilovich, Gade, Pippel, Hoffmeyer, Tchesunov, Stevens, Winkler, Hughes, Traikov, Hiller, Rivkina, Schiffer, Myers & Kurzchalia **sp. nov**.
urn:lsid:zoobank.org:act:57A9E39B-5603-46B6-A035-4B9BDBC1C441

### Discovery site and radiocarbon dating

Previously, we had shown that nematodes from the Siberian permafrost with morphologies consistent with the genera *Panagrolaimus* and *Plectus* could be reanimated thousands of years after they had been frozen. Several viable nematode individuals were found in two of the more than 300 studied samples of permafrost deposits spanning different ages and genesis. Samples were collected by researchers of the Soil Cryology Lab, Pushchino, Russia, during perennial paleo-ecological expeditions carried out in the coastal sector of the northeastern Arctic [12]. The detailed description of the study site (outcrop Duvanny Yar, Kolyma River, Fig 1A), sampling and revitalizing procedures are provided in S1 Text. Like other late Pleistocene permafrost formations in the northeastern Arctic, Duvanny Yar is composed of permanently frozen ice-rich silt deposits riddled with large polygonal ice wedges that divide them into mineral blocks [29,30] (Fig 1B). Sediments include sandy alluvial layers, peat lenses, buried paleosols and Pleistocene rodent burrows (Fig 1C). The burrow (P-1320), in which *Panagrolaimus* nematodes were found (Fig 1D), has been taken from the frozen outcrop wall at a depth of about 40 m below the surface and about 11 m above river water level in undisturbed and never thawed late Pleistocene permafrost deposits. The fossil burrow left by arctic gophers of the genus *Citellus* consists of an entrance tunnel and large nesting chamber up to 25 cm in diameter [29].

The sterility of permafrost sampling and age of cultivated biota have been discussed in detail in several reviews [9,31,32]. Based on previous reports, the age of the organisms found in a burrow is equal to the freezing time and corresponds to the age of organic matter conserved in the syncryogenic sediments. This makes it possible to use radiocarbon dating of organic matter to establish the age of organisms. We performed Accelerator Mass Spectrometry (AMS) radiocarbon analysis of plant material obtained from studied borrow P-1320 and determined a direct $^{14}$C age of 44,315±405 BP (Institute of Geography, RAS; sample IGAN$_{AMS}$ 9137). Calibrated age range is 45,839–47,769 cal BP (95.4% probability) (S1 Fig).

### Like other parthenogenetic *Panagrolaimus*, the newly discovered species is triploid

The revived animal was cultivated in the laboratory for over 100 generations and initially described as *Panagrolaimus* aff. *detritophagus* [33] based on morphology. We conducted a detailed morphological analysis of the revived animal (Figs 2 and S2, and Table A in S1 Text

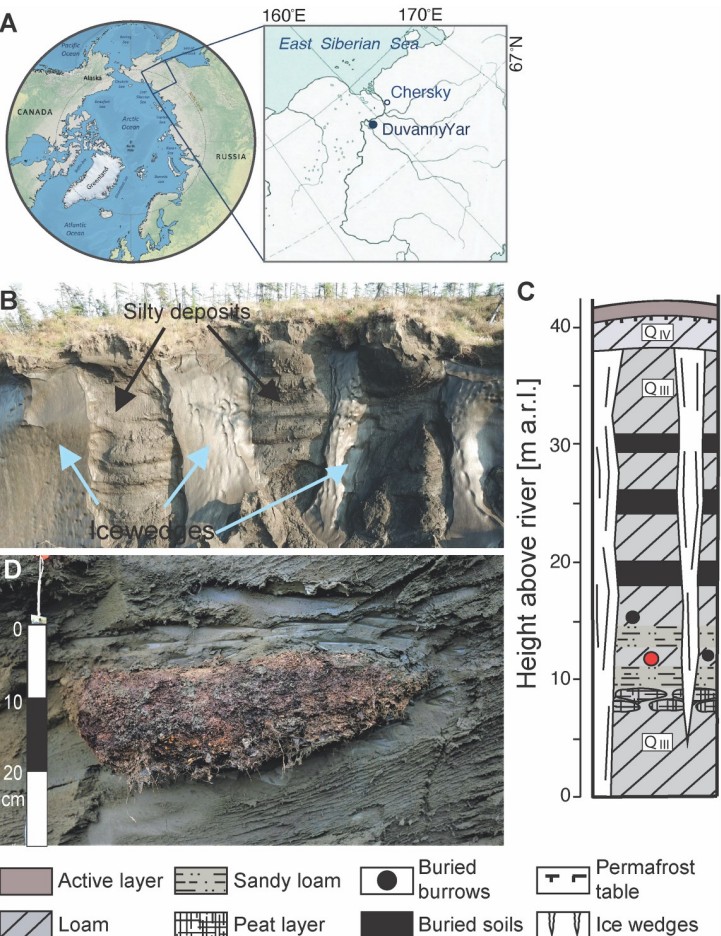

**Fig 1. Study site.** (A) Location of the Duvanny Yar outcrop on the Kolyma River, northeastern Siberia, Russia. https://climate.copernicus.eu/sites/default/files/inline-images/C3S_indicator_sea_ice_sidebar_figS2_branded.png) (https://climate.copernicus.eu/data-protection-and-privacy-statement) (B) View of the upper part of outcrop composed of ice wedges and permafrost silty deposits. (C) Lithostratigraphic scheme of deposits, showing location of studied rodent borrow (red circle). (D) fossil rodent burrow with herbaceous litter and seeds buried in permafrost deposits; m a.r.l. = meters above river level.

and Box 1), which confirmed unambiguously that the animal belongs to the genus of *Panagrolaimus*, in agreement with a previous phylogenetic analysis of the 18S ribosomal RNA sequence [33]. However, due to the morphological uniformity of *Panagrolaimus*, unusual even for nematodes, morphology and molecular analysis of a single ribosomal RNA sequence is insufficient to describe a species. We found the species to be parthenogenetic, which further complicates description under most species concepts. Due to these limitations, we decided to refer to the phylogenetic species concept, using phylogenetic trees based on multiple genes as markers.

To obtain comprehensive molecular data for phylogenomic species determination, we generated a genome assembly using PacBio HiFi sequencing with long reads (84X coverage, mean length 14,425 bp). Our analysis of repeat and gene content is described in Table C in S1 Text. K-mer analysis of the reads clearly indicated that this animal has a triploid genome (Fig 3A), like other parthenogenetic *Panagrolaimus* species [34]. Despite the challenges that a triploid genome poses for assembly, we obtained a highly contiguous contig assembly of the three pseudohaplotypes that comprise almost 266 Mb and thus have a similar genome size as other

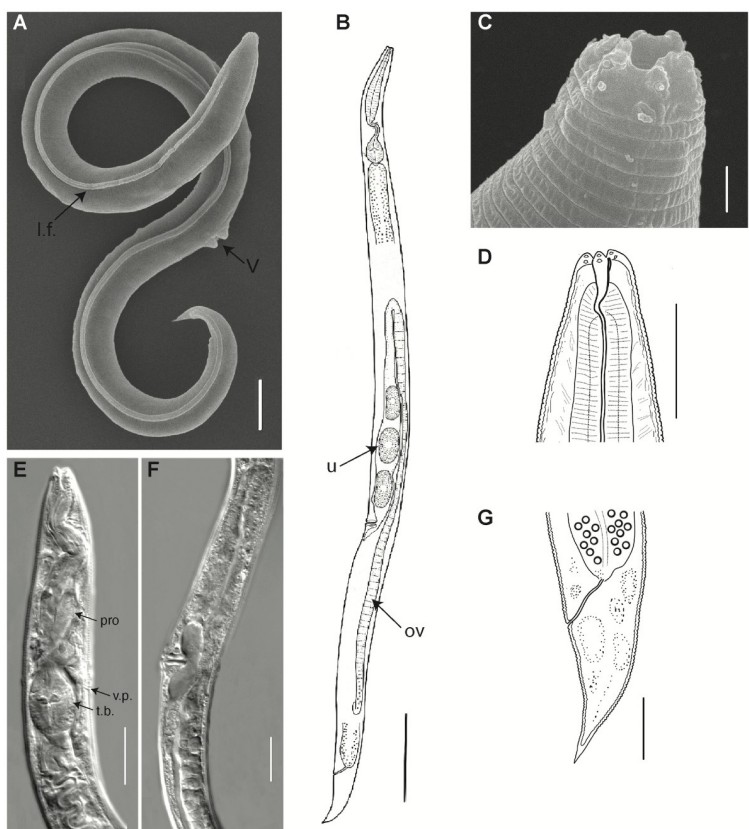

**Fig 2. General morphology of *P. kolymaensis*, female.** Scanning electron pictures (A, C), light microscopy photographs (E, F) and graphic presentations (B, D, G) of holotype: A, B) entire body, C, D) anterior ends, E) anterior body, F) perivulvar body region, G) tail. Abbreviations: l.f.–lateral field, ov–ovary, pro–procorpus of the pharynx, t.b.–terminal bulb of the pharynx, u–uterus with eggs, v–vulva, v.p.–ventral pore. Scale bars: A, D, E, F, G– 20 μm, B–100 μm, C– 2 μm.

parthenogenetic *Panagrolaimus* species [34]. The contig N50 value of all three pseudohaplotypes is 3.8 Mb. Since these pseudohaplotypes exhibited a noticeable degree of divergence, we further investigated their relationship by using the apparent homeologs in our gene predictions to align the longest continuous contigs based on micro synteny (Fig 3B). Links between the contigs clearly show the triploid state of the genome.

## Applying a phylogenomic concept to define *P. kolymaensis*

To place the species in the genus *Panagrolaimus*, we conducted a broader multi-gene phylogenomic analysis using Maximum likelihood methods. Our analysis of a concatenated, partitioned alignment of 60 genes, and a coalescence-based approach using a broader set of 12,295 gene trees, retrieves the revived animal as sister to all other sequenced *Panagrolaimus* species, but as an ingroup to *Propanagrolaimus* [35] (Figs 3C, S3B, and S3C). Thus, the phylogenetic placement provides strong evidence that this animal represents a novel species. Furthermore, there is substantial sequence divergence between this novel species, and *Panagrolaimus* sp. PS1159 and *Panagrolaimus* sp. ES5, estimated to be on average 2.06 and 2.11 amino acid substitutions per site in our concatenated alignment, respectively. The substantial divergence is in line with previous data on ages of *Panagrolaimus* nematodes [34], and more broadly seen in nematodes, which can be hyper-diverse [36,37]. Our data also contradicts the assumption that

## Box 1. Description of *P. kolymaensis*

| | |
|---|---|
| **Description** | Body spindle-shaped and usually almost straight after fixation (Fig 2A and 2B). Cuticle thin and faintly annulated, 10–11 annules per 10 μm in cervical region, 13–14 in midbody, preanally again 10 annules within 10 μm. Conspicuous convex lateral fields with three incisures 1.5–2 μm wide extended along the body from about ¼–1/3 procorpus length to 2/3 of tail length. In SEM, the lateral field looks like a bolster with a narrow median split. Labial region set off. Mouth opening surrounded with six lips (Fig 2C and 2D). Anterior sensilla as papillae arranged in two close but separate subsequent circles. Somatic sensilla (i.e., deirids and phasmids) not evident. Buccal cavity cylindro-conoid, and unarmed; its total length 9–13 μm, maximum stoma width 1.7–2.8 μm (Fig 2E). Dorsal stoma wall (dorsal rhabdion) more clearly sclerotized. Anterior part of the buccal cavity comprising cheilostom and gymnostom nearly cylindroids while stegostom conically narrowed and ended with a distinct tight flexion. Pharynx consists of three distinct parts: straight anterior procorpus, narrow medial isthmus and rounded terminal bulb. Procorpus gradually widening to it posterior end, always straight in all specimens, with transversal muscular striation, more prominent in posterior three fourths. Isthmus narrow, cylindroid, bent starkly in nearly all studied specimens. Terminal bulb strongly muscular, with a valvular apparatus at about 40% of the bulb length. Cardia in shape of truncate cone. Intestine (midgut) tissue filled with vacuoles and granules; in the anterior most region, the granules smaller and look more pallid. Cell borders not visible in the intestine, but internal lumen distinct, sinusoid. No internal content visible in the internal lumen.<br><br>Ventral excretory-secretory pore and its cuticularized duct situated at the level of anterior part of the bulb. No other details of the excretory-secretory system visible. Vulval lips protruding. Genital branch monodelphic prodelphic and situated dorsally and to the right of the midgut (Fig 2F). Vagina distinctly cuticularized and opens to the elongate uterus. A long oviduct extended anterial from the anterior uterus; the oviduct folded up and then posterior ward and transforms into an elongate ovary. There are one or two ripe eggs in the uterus in most specimens. Tail short conical, with short acute spike-like mucro (Fig 2G). |
| **Etymology** | Species name *kolymaensis* (Latin) is derived from the Kolyma River area. |
| **Holotype** | Senckenberg Natural History Museum, Frankfurt am Main, Germany (collection number SMF 17067). |
| **Paratypes** | Senckenberg Natural History Museum, Frankfurt am Main, Germany (collection numbers SMF 17068, SMF 17069) (eighteen paratypes). |
| **Type locality** | Frozen fossil rodent burrow buried in permafrost 45,839–47,769 cal BP, 40 meters from the surface, outcrop Duvanny yar, Kolyma River, North-East of Siberia, Russia (68.633410, 159.078800). Frozen material from burrow was collected by Dr.S.Gubin (Soil Cryology Lab, Pushchino, Russia) in august 2002. |

all parthenogenetic strains in the genus *Panagrolaimus* have a monophyletic origin [34] (Fig 3C). Using the gene-tree reconciliation approach implemented in GRAMPA (Gene-tree Reconciliation Algorithm with MUL (Multi labelled)-trees for Polyploid Analysis), we explored whether the additional set of proteins we found in the parthenogenetic species is a result of auto- or allopolyploidy. Finding the additional proteins basally branching, outside of the lineage containing both parthenogenetic and sexual species, suggests that an allopolyploid origin of these extra gene-copies (S6 Fig).

Based on the Kolyma River location where the animal was unearthed, we propose the following taxonomic classification and species name:

Phylum Nematoda Potts, 1932

Class Chromadorea Inglis 1983

Suborder Tylenchina Thorne, 1949

Family Panagrolaimidae Thorne, 1937

*Panagrolaimus kolymaensis*

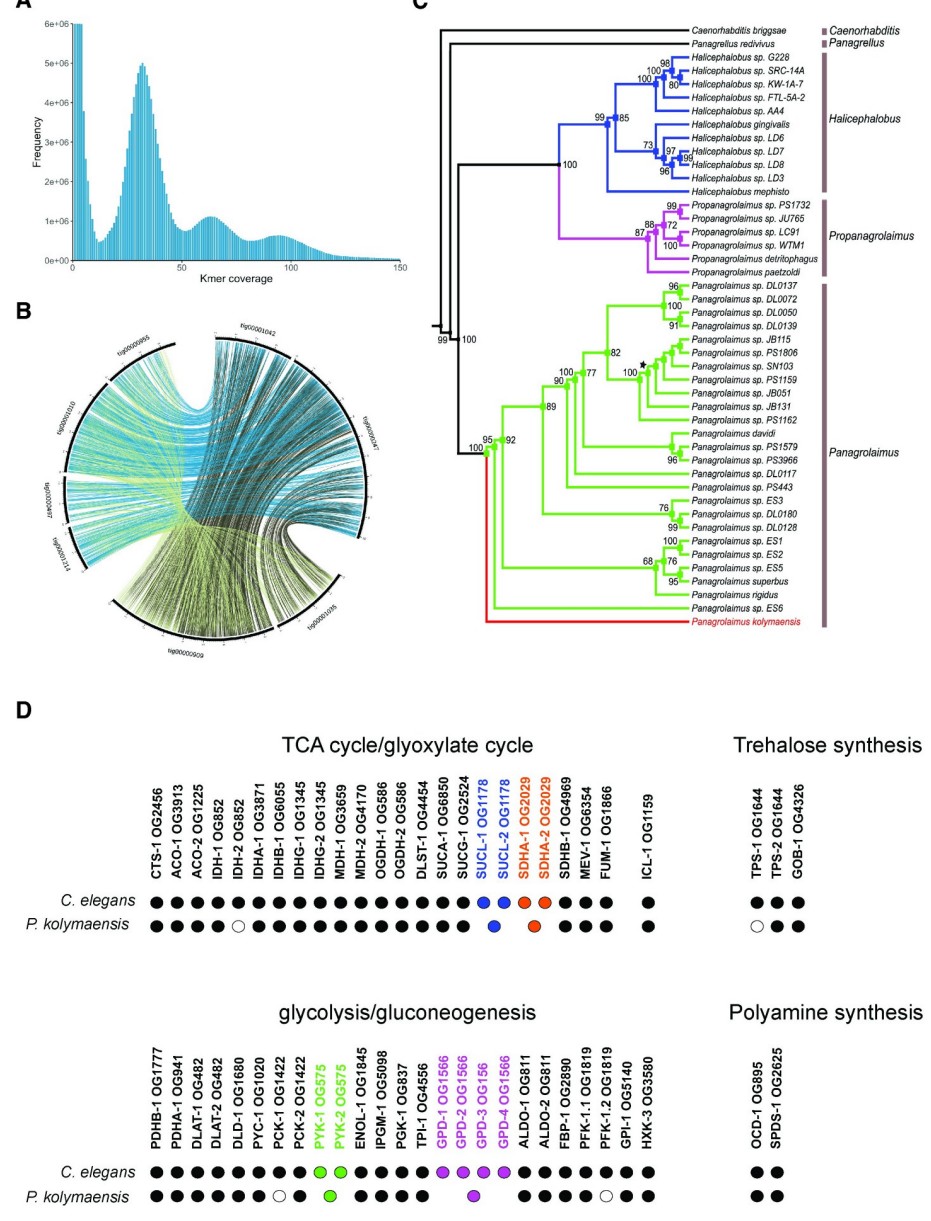

**Fig 3. Genome assembly and phylogenomics reveals that the newly discovered *P. kolymaensis* species is triploid.**
A) Kmer spectra of the *P. kolymaensis* PacBio HiFi data. Kmers of length 19 were counted using Jellyfish. B) Circos plot showing the triploid structure of the *P. kolymaensis* genome. Lines represent the position of 6,715 homeologs in eight contigs that comprise 39.9 Mb (15%) of the assembly. Homeologs were identified by clustering protein-coding genes into orthogroups using OrthoFinder and selecting groups containing three sequences. Contig IDs and scale is shown. C) Inferred species tree for all taxa. The maximum likelihood tree inferred using a concatenated supermatrix (18S and 28S genes) with bootstrap support values is displayed. All genera are represented as monophyletic clades. *P. kolymaensis* is highlighted in red and basal to all other *Panagrolaimus* taxa. Internal nodes, where all subsequent branches represent identical sequences, are displayed with a black star. D) *P. kolymaensis* possesses *C. elegans* gene orthologs to enzymes required for TCA cycle, glyoxylate shunt, glycolysis, gluconeogenesis, trehalose synthesis, and polyamine synthesis. Black filled circles: Ortholog presence suggested by orthogroup clustering, phylogenetic analysis, and domain architecture. White filled circles: No ortholog found via current analysis. Coloured filled circles: Presence of *P. kolymaensis* gene(s), related to several *C. elegans* genes (all genes of same colour) that are all co-orthologous to that gene (those genes). Label: *C. elegans* enzyme names and orthogroup that contains that gene according to our orthogroup clustering.

### *C. elegans* dauer larvae and *P. kolymaensis* might utilize partially similar mechanisms to enter and remain in cryptobiotic state for prolonged periods of time

In the absence of established genetic methods in *P. kolymaensis*, we referred to the model *C. elegans* as a comparator system to gain insights into possible pathways for long term survival [25,27,28,33]. The high-quality genome of *P. kolymaensis* allowed us to compare its molecular toolkit for cryptobiosis with that of *C. elegans*. We used orthology clustering and phylogenetics to investigate whether the genome of *P. kolymaensis* contains genes previously implicated in cryptobiosis in the *C. elegans* dauer larva. Our analysis showed that, like other *Panagrolaimus* species [38,39], *P. kolymaensis* also encodes orthologs to a *C. elegans* trehalose phosphate synthase gene (*tps-2*) and to a trehalose phosphatase gene (*gob-1*) (Fig 3D and S1 Orthology analysis). Furthermore, we found orthologs to all *C. elegans* enzymes required for polyamine biosynthesis, the TCA cycle, glycolysis, gluconeogenesis, and glyoxylate shunt (Fig 3D and S1 Orthology analysis) suggesting that *P. kolymaensis* might partially utilize similar molecular mechanisms as *C. elegans* to facilitate survival of unfavorable conditions.

Our earlier findings established that amongst several developmental stages of *C. elegans*, only the dauer larva, formed during unfavorable conditions (such as low nutrients and high population density), could survive anhydrobiosis and exposure to freezing [25,33]. The dauer larva is in a hypometabolic state with distinct metabolic properties such as reduced oxygen consumption and heat dissipation in comparison to other larval stages of *C. elegans*. To survive extreme desiccation, *C. elegans* dauer larvae (in its hypometabolic state) need to be first preconditioned at high relative humidity (98% RH) for 4 days [25]. During preconditioning, dauer larvae upregulate trehalose biosynthesis that ensures their survival to harsh desiccation [25,28]. We tested whether the survival of *P. kolymaensis* is also facilitated by preconditioning. As there is no dauer stage in the *Panagrolaimus* life cycle, we performed our experiments with a population of all the larval stages and adults. Although a small proportion of *P. kolymaensis* individuals survive harsh desiccation and freezing without preconditioning (Fig 4A), the mixture of all the larval stages and adults of *P. kolymaensis* survive significantly higher (p value< 0.0001) in proportion to harsh desiccation upon preconditioning (Fig 4A). Similarly, preconditioning and desiccation further enhanced survival rate of *P. kolymaensis* to freezing (-80˚C) (Fig 4A). Like *C. elegans* dauer larva, *P. kolymaensis* upregulates trehalose levels up to 20-fold upon preconditioning (Fig 4B). We previously reported that to upregulate trehalose levels upon preconditioning, *C. elegans* dauer larva dissipate their fat reserves (Triacylglycerols) by activating the glyoxylate shunt and gluconeogenic pathway [28] Upon preconditioning, we found that triacylglyceride (TAG) levels are significantly decreased in *P. kolymaensis* (S5A and S5B Fig). To further investigate whether acetyl-CoA derived from the degradation of TAGs culminates in trehalose, we applied the previously developed method of metabolic labelling with [14]C-acetate in combination with 2D-TLC [1,33]. The [14]C-acetate metabolized by the worms is incorporated into TAGs. Upon degradation of TAGs, [14]C-acetyl CoA is released which acts as a precursor for trehalose biosynthesis. As shown in Fig 4C, preconditioning led to a huge increase of radioactivity in trehalose and to a small increase in some amino acids (glycine/serine, phenylalanine; panels C and D). Interestingly, *P. kolymaensis* displayed an additional spot (Fig 4D, enumerated as 7), that was not found in *C. elegans*. We identified this spot as trehalose-6-phosphate (S5C–S5H Fig), a precursor of trehalose, based on the fragmentation pattern of the molecule, using mass spectrometry. Thus, to resist harsh desiccation, like *C. elegans* dauer larvae, *P. kolymaensis* might utilize the glyoxylate shunt and consequently acetate derived from TAGs to synthesize trehalose. Detection of the immediate precursor (trehalose-6-phosphate) suggests that the flux of metabolites is intense in *P. kolymaensis*. Finally, we

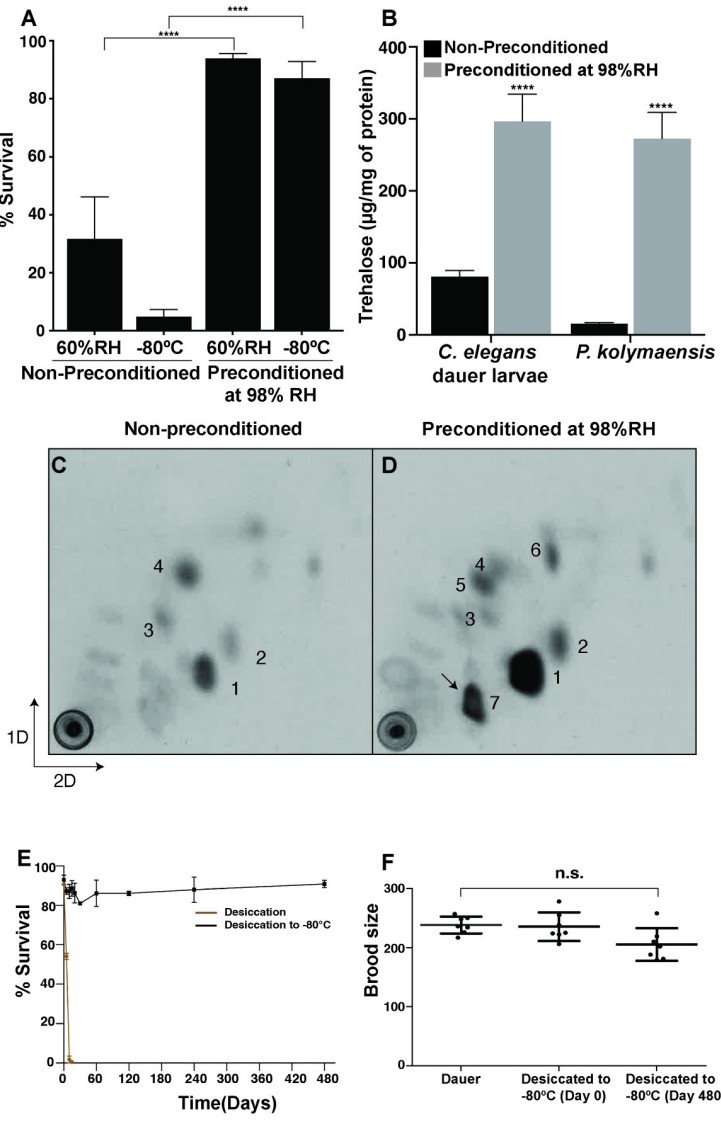

**Fig 4. *C. elegans* dauer larvae and *P. kolymaensis* might utilize similar mechanisms to survive cryptobiosis.** A) Survival rate of *P. kolymaensis* nematodes to desiccation and freezing (-80˚C). Error bars indicate standard error of mean of two independent experiments with two technical replicates performed on two different days. Statistical comparison was performed using unpaired t-test with Welch correction. n.s. p > 0.05, ****p < 0.0001. For desiccation (non-preconditioned) n = 289, freezing (non-preconditioned) n = 675, desiccation (preconditioned at 98%RH) n = 953 and freezing (preconditioned at 98%RH) n = 1295. B) *P. kolymaensis* nematodes and *daf-2(e1370)* dauer larvae upregulate trehalose levels upon preconditioning at 98%RH. Error bars indicate standard error of mean of two independent experiments with three replicates performed on two different days. Statistical comparison was performed using two-way ANOVA with Holm-Sidak's multiple comparison test, ****p < 0.0001. C-D) 2D-thin layer chromatography of $^{14}$C-acetate labelled metabolites from *P. kolymaensis* that were non-preconditioned and preconditioned at 98%RH. Enumerated spots indicate trehalose (1), glucose (2), glutamate (3), glutamine (4), serine/glycine (5) and phenylalanine (6). Representative images from at least two independent experiments performed on two different days. E) Desiccated *daf-2 (e1370)* dauer larvae survive to freezing (-80˚C) for an extremely long period. Error bars indicate standard error of mean of two independent experiments with two technical replicates performed on two different days. F) Brood size of desiccated dauer larvae exposed to freezing remain like that non- desiccated dauer larvae. Average brood size is the mean of seven dauer larvae per each condition. Statistical comparison was performed by using non-parametric Kolmogorov-smirnov test n.s p>0.05.

investigated whether *C. elegans* dauer larvae can also survive in prolonged cryptobiotic state. Despite preconditioning, the survival ability of desiccated dauer larvae at room temperature declines very rapidly, with most larvae dead after almost 10 days (Fig 4E) (S4A and S4B Fig). Direct freezing without any cryoprotectants at -80 ˚C leads to instant death of the animals. To test whether combining these conditions could extend the viability of dauer larvae (S4A and S4B Fig), we transferred the desiccated larvae to -80 ˚C. Remarkably, under these conditions, there was no significant decline in viability even after 480 days (Fig 4E). Moreover, after thawing the animals resumed reproductive growth and produced progeny in numbers like those of animals kept under control conditions (Fig 4F). Since we did not observe any reduction in the survival at any time points, this suggests that the combination of anhydrobiosis and freezing can prolong the survival ability of dauer larvae. Thus, *C. elegans* dauer larvae, when exposed to combination of cryptobiotic states can survive for extremely long periods of time.

## Discussion

The new nematode species from permafrost can now be placed into the genus *Panagrolaimus* [40], which contains several described parthenogenetic and gonochoristic species [34,41]. Many *Panagrolaimus* display adaptation to survival in harsh environments [22] and the genus includes the Antarctic species *P. davidi* [23]. The genus *Panagrolaimus* is exceptional in its morphological uniformity even among nematode species that are hard to classify based on morphology in general. Thus, species designation via microscopic (including SEM) analysis is unreliable, which is further complicated by the absence of males in parthenogenetic species. Males have an important diagnostic feature such as spicules and pericloacal papillae, females differ from one species to another mainly by morphometrics, where interspecies differences (absolute measures and ratios) might be subtle. Our specimens are similar based on absolute sizes and ratios to females of the bisexual species *Panagrolaimus detritophagus* [42]. The only non-overlapping morphometric character is index "b" (body length: pharynx length): 5.6–6.8 in *P. kolymaensis* versus 4.4–5.1 in *P. detritophagus*.

Consequently, we turned to phylogenomic methods under the phylogenetic species concept to place the species on the tree. This showed that this species is an outgroup to other known *Panagrolaimus* species, raising the possibility of a second independent evolution of parthenogenesis in the genus, in contrast to previous findings [34,35,41]. Alternatively, the hybrid origin of parthenogenetic *Panagrolaimus* could influence the phylogenetic positioning of strains, raising the possibility that the new species is a true sister to the other parthenogenetic strains. To fully resolve the phylogenetic positioning further, extensive sampling, and genome sequencing of *Panagrolaimus* species is needed. We found *P. kolymaensis* to be triploid and thus a hybrid origin is possible, as seen in other parthenogenetic *Pangrolaimus* [34].

The highly contiguous genome of *P. kolymaensis* will allow for analyses of this trait in comparison to other *Panagrolaimus* species currently being genome sequenced. Our results provide a deeper insight into the homology of molecular and biochemical mechanisms between *C. elegans* and *P. kolymaensis*, which are not only taxonomically but also ecologically distinct. *C. elegans* can mostly be found in rotting fruits and plants in temperate regions [43,44], while *Panagrolaimus* species are globally distributed and prevalent in leaf litter and soil [41], including in harsh environments [22].

We show through orthology analysis that the well-studied molecular pathways used by *C. elegans* larvae to enter the dauer state, such as insulin [45,46] (DAF-11, DAF-2 & DAF-16), TGF-β [47] (DAF-7), steroid [48] (DAF-9, DAF-12) are present in the genome of the *P. kolymaensis* (S4C Fig). The presence of homologous genes in two species does not necessarily

demonstrate their functionality in both. Therefore, further functional analyses are needed to study molecular pathways in detail. Trehalose accumulation (Fig 4B) and depletion of triacyl-glycerols (S5A and S5B Fig) ensure the functionality of the trehalose biosynthesis pathway and utilization of glyoxylate shunt during desiccation in *P. kolymaensis*. Without the activity of the enzyme TPS-2 and glyoxylate shunt, it is unfeasible to synthesize trehalose in nematodes. We do not eliminate the possibility of other biochemical features that might contribute to the des-iccation survival ability of *P. kolymaensis*, but with regards to trehalose biosynthesis and the glyoxylate shunt, our data suggest that the molecular tool kit is partially orthologous. In our future studies, we intend to perform RNAi-based experiments to infer the concrete mecha-nisms. Our results hint at convergence or parallelism in the molecular mechanisms organizing dauer formation and cryptobiosis.

As mentioned above, preconditioning enhances the survival of *P. kolymaensis* by rendering them desiccation tolerant. We previously reported that preconditioning elevates trehalose bio-synthesis in *C. elegans* dauer larvae and the elevated trehalose renders desiccation tolerance by protecting the cellular membranes [25]. It is not surprising that *P. kolymaensis* upregulates tre-halose, however the magnitude of trehalose elevation is higher than *C. elegans* dauer larvae. This indicates that central regulators (DAF-16, DAF-12) of trehalose upregulation may differ-entially regulate *tps-2* in *P. kolymaensis* [38,49,50]. Although *P. kolymaensis* utilizes the glyoxy-late shunt and gluconeogenesis to upregulate trehalose levels, it is intriguing to observe that they accumulate substantial levels of trehalose-6-phosphate. Further investigation of this observation using RNAi or inhibitor-based experiments will provide insights into molecular mechanisms of metabolic regulation in *P. kolymaensis* upon preconditioning. Our findings for the first time demonstrate that *C. elegans* dauer larvae possess an inherent ability to survive freezing for prolonged periods if they undergo anhydrobiosis. It is tempting to speculate that undergoing anhydrobiosis might be a survival strategy of *C. elegans* to survive the seasonal changes in nature.

In summary, our findings indicate that by adapting to survive cryptobiotic state for short time frames in environments like permafrost, some nematode species gained the potential for individual worms to remain in the state for geological timeframes. This raises the question of whether there is an upper limit to the length of time an individual can remain in the cryptobio-tic state. Long timespans may be limited only by drastic changes to the environment such as strong fluctuations in ambient temperature, natural radioactivity, or other abiotic factors. These findings have implications for our understanding of evolutionary processes, as genera-tion times may be stretched from days to millennia, and long-term survival of individuals of species can lead to the refoundation of otherwise extinct lineages. This is particularly interest-ing in the case of parthenogenetic species, as each individual can find a new population with-out the need for mate finding, i.e. evading the cost of sex. Finally, understanding the precise mechanisms of long-term cryptobiosis and cues that lead to successful revivals can inform new methods for long term storage of cells and tissues.

## Methods

### Nomenclatural Acts

The electronic edition of this article conforms to the requirements of the amended Interna-tional Code of Zoological Nomenclature, and hence the new names contained herein are avail-able under that Code from the electronic edition of this article. This published work and the nomenclatural acts it contains have been registered in ZooBank, the online registration system for the ICZN. The ZooBank LSIDs (Life Science Identifiers) can be resolved and the associated information viewed through any standard web browser by appending the LSID to the prefix

## Materials and *C. elegans* strains

[1-14C] -acetate (sodium salt) was purchased from Hartmann Analytic (Braunschweig, Germany). All other chemicals were purchased from Sigma-Aldrich (Taufkirchen, Germany). The *Caenorhabditis* Genetic Centre (CGC) which is funded by NIH Office of Research Infrastructure Programs (P40 OD010440) provided *daf-2(e1370)* and *E. coli NA22* strains.

## Genomic DNA isolation from *P. kolymaensis* nematodes

After isolation (S1 Text), to ensure our strain *P. kolymaensis* (Pn2-1) can adapt to different laboratories, we grew them for multiple generations. The strain was in culture for several generations during genomic DNA isolation and was frozen after genomic DNA isolation was performed. *P. kolymaensis* nematodes (isofemale strain Pn2-1) were grown on several plates of NGM agar plated with *E. coli NA22* bacteria at 20˚C. Worms were collected from the plates, washed with water at least three to five times by centrifugation at 1000 g to remove any residual bacteria and any debris. The worm pellet was dissolved in 5 volumes of worm lysis buffer (0.1M Tris-HCl pH = 8.5, 0.1M NaCl, 50mM EDTA pH = 8.0) and distributed in 1.5 ml of microcentrifuge tubes. These tubes are incubated at -80˚C for 20 minutes. 100 µl of Proteinase 'K' (20 mg/ml) was added to each tube and they are incubated at 60˚C overnight. 625 µl of cold GTC buffer (4M Guanidinium Thiocynate, 25mM Sodium citrate, 0.5% (v\v) N-lauroylsarcosine, 7%(v/v) Beta Mercaptoethanol) was added to the tube, incubated on ice 30 min, and mixed by inverting every 10 min. 1 volume of phenol–chloroform-isoamyl alcohol (pH = 8) was added to the lysate and mixed by inverting the tube 10–15 times. Tubes were centrifuged for 5 min at 10,000 g at 4˚C to separate the phases. The upper aqueous phase was carefully collected into a fresh tube. One volume of fresh chloroform was added and mixed by inverting the tubes for 10–15 times and centrifuged for 5 min at 10,000 g at 4˚C to separate the phases. One volume of cold 5 M NaCl was added, mixed by inverting the tubes and incubated on ice for 15 min. After incubation these tubes were centrifuged for 15 min at 12,000–16,000 g at 4˚C. The supernatant containing the nucleic acids were slowly transferred into a fresh tube. One volume of isopropanol was added to the tube, inverted few times, and incubated on ice for 30 minutes. After incubation, the tubes were centrifuged at 3000 g for 30–45 min at 25˚C and the supernatant was discarded without disturbing the pellet. The pellet was washed twice with 1 ml of 70% ethanol, tubes were centrifuged at 3000 g for 5 min and supernatant was discarded and incubated at 37˚C for 10–15 min to dry the pellet. The pellet was resuspended carefully in TE buffer. The quality of the genomic DNA was analyzed with pulse field gel electrophoresis.

## Genome sequencing and assembly

The long insert library was prepared as recommended by Pacific Biosciences according to the 'Procedure & Checklist-Preparing gDNA Libraries Using the SMRTbellExpress Template Preparation Kit 2.0' protocol. In summary, RNAse treated HMW gDNA was sheared to 20 kb fragments on the MegaRuptordevice (Diagenode) and 10 µg sheared gDNA was used for library preparation. The PacBio SMRTbell library was size selected in two fractions (9-13kb, > 13kb) using the BluePippin device with cassette definition of 0.75% DF MarkerS1 3–10 kb

Improved Recovery. The second fraction of the size-selected library was loaded with 95 *p*M on plate on a Sequel SMRT cell (8M). Sequel polymerase 2.0 was used in combination with the v2 PacBio sequencing primer and the Sequel sequencing kit 2.0EA, with a runtime of 30 hours. We created PacBio CCS reads from the subreads.bam file using PacBio's ccs command linetool (version4.2.0), outputting 8.5Gb of high-quality CCS reads (HiFi reads N50 of 14.4 kb). HiCanu (version 2.2) [51] was used to create the contig assembly. Blobtools [52] (version 1.1.1) was used to identify and remove bacterial contigs. The final triploid contig assembly consists of 856 contigs has a N50 of 3.82 Mb and a size of 266Mb. The mitochondrial genome was created with the mitoHifi pipeline (version 2, https://github.com/marcelauliano/MitoHiFi based on the assembled contigs and the closely related reference mitochondrial genome of *Panagrellus redivivus* (strain: PS2298/MT8872, ENAaccession: AP017464). The mitoHifi pipeline identified 49 mitochondrial contigs ranging from 13-32Kb. The final annotated circular mitochondrial genome has a length of 17467 bp.

To identify pseudohaplotypes in the *P. kolymaensis* genome assembly, we selected the longest isoform of each predicted protein-coding gene in our assembly and in the *C. elegans* genome (downloaded from WormBase Parasite, release WBPS15) using AGAT (version 0.4.0) and clustered them into orthologous groups (OGs) using OrthoFinder (version 2.5.2). We identified OGs that contained three *Panagrolaimus* sequences (i.e. groups that were present as single-copy in all three pseudohaplotypes) and used these to identify trios of multi-megabase size contigs derived from the three pseudohaplotypes. Synteny between the three pseudohaplotypes was visualized using Circos to plot the positions of each homeolog (version 0.69–8).

## Genome annotation

RepeatModeler 1.0.8 (http://www.repeatmasker.org/) was used with parameter '-engine ncbi' to create a library of repeat families which was used with RepeatMasker 4.0.9 to soft-mask the *Panagrolaimus* genome. To annotate genes, we cross mapped protein models from an existing *Panagrolaimus* as external evidence in the Augustus based pipeline. The completeness of our predictions was evaluated using BUSCO on the gVolante web interface.

## Orthology analysis

We conducted a gene orthology analysis using genomic data from *P. kolymaensis*, the Plectid nematode species from the permafrost, as well as genomic data from WormBase Parasite (https://parasite.wormbase.org; accessed 17/12/2020): *Caenorhabditis elegans*, *Diploscapter coronatus*, *Diploscapter pachys*, *Halicephalobus mephisto*, *Panagrellus redivivus*, *Panagrolaimus davidi*, *Panagrolaimus* sp. ES5, *Panagrolaimus* sp. PS1159, *Panagrolaimus superbus*, *Plectus sambesii*, and *Propanagrolaimus* sp. JU765. For plectids, genomic resources are scarce. We therefore added transcriptome data of *Plectus murrayi*, *Anaplectus granulosus*, *Neocamacolaimus parasiticus*, and *Stephanolaimus elegans*, with the latter three transcriptomes kindly provided by Dr. Oleksandr Holovachov (Swedish museum of natural history). Transcriptomes for *Anaplectus granulosus*, and *Neocamacolaimus parasiticus* have been published and are readily available [53,54]. All three transcriptomes were assembled *de novo* with Trinity [55]. The exact procedures are described in the respective publications [53,54]. The *Stephanolaimus elegans* transcriptome was assembled using the same methodologies as *Neocamacolaimus parasiticus*.

The *Plectus murrayi* transcriptome was built from raw reads deposited at NCBI (https://sra-downloadb.be-md.ncbi.nlm.nih.gov/sos2/sra-pub-run-13/SRR6827978/SRR6827978.1; accessed 22.12.2020) and assembled using Galaxy Trinity version 2.9.1 [55,56]. All default options were used including *in silico* normalization of reads before assembly. Transdecoder (conda version 5.5.0) [57] was used to translate to amino acid sequence. Identical reads were

removed with cd-hit version 4.8.1 [58,59], with shorter isoforms removed using the Trinity get_longest_isoform_seq_per_trinity_gene.pl command [57] (Trinity conda version 2.8.5; Anaconda Software Distribution, Conda, Version 4.9.2, Anaconda, Nov. 2020). Amino acid translations of the longest isoforms were extracted with AGAT (Dainat, https://www.doi.org/10.5281/zenodo.3552717) from genome assembly FASTA files and genome annotation GFF3 files using the 'agat_convert_sp_gxf2gxf.pl', 'agat_sp_keep_longest_isoform.pl' and 'agat_-sp_extract_sequences.pl' scripts, respectively. All FASTA headers were modified to allow for simple species assignment of each sequence in subsequent analysis. Orthology analysis was conducted with OrthoFinder v. 2.5.1 [60,61] using default settings. For genes of interest, we constructed alignments with MAFFT v. 7.475 [62] using the localpair and maxiterate (1000) functions. Spurious sequences and areas that were not well aligned were removed with Trimal v. 1.4.rev22 [63] (procedure stated in S1 Orthology analysis below each phylogeny). We then ran phylogenetic analysis with Iqtree2 v. 2.0.6 [64], with -bb 1000 option, testing the model for each analysis (models eventually used stated in S1 Orthology analysis). PFAM domains were explored using Interproscan v. 5.50–84.0 [65]. The phylogenies were visualized with Dendro-scope 3.7.6 [66] and figures were created with Inkscape (https://inkscape.org). Most of our analysis was performed on the HPC RRZK CHEOPS of the Regional Computing Centre (RRZK) of the University of Cologne.

## Phylogenomics

Sequences of 18S and 28S genes from 44 taxa across the *Propanagrolaimus*, *Panagrolaimus*, *Panagrellus* and *Halicephalobus* genera (all listed in S1 Text were aligned (MAFFT L-INS-I v7.475) [62], concatenated [67]and used to infer a species tree using maximum likelihood via (IQTREE) [68] and partitioned by best-fit models of sequence evolution for both [69]. Nodal support was determined using 1000 bootstrap pseudoreplicates. A further 60 genes from 101 taxa (all listed in S1 Text) were used to confirm the taxonomic position using the supermatrix concatenation methods outlined above. Given the limitations of differential gene sampling, we expanded our phylogenomic analyses to include a coalescence approach using 12,295 ML gene trees inferred for orthogroups containing the target animal. Instances of multiple genes per species per group were treated as paralogs/orthologs and analysed using ASTRAL-Pro [70]. Given the number of copies of genes per orthogroup, we explored whether auto or alloploidy was the source of extra genes observed using the gene-tree reconciliation approach implemented in GRAMPA (Gene-tree Reconciliation Algorithm with MUL (Multi labelled)-trees for Polyploid Analysis) [71]. All gene trees rooted at the midpoint and the final ASTRAL-pro species tree were used as inputs, with the most parsimonious result analyzed further.

## Desiccation survival assay

*C. elegans* dauer larvae desiccation assays were performed as described in [25]. *P. kolymaensis* desiccation assays were performed similarly as described in [25] with mixed population (Mixture of all larval stages and adults) of the nematodes.

## Exposure of nematodes to extreme environments

*C. elegans* dauer larvae or mixed population (Mixture of all larval stages and adults) of *P. kolymaensis* nematodes were preconditioned and desiccated as described in [28], then transferred to elevated temperature of 34˚C, freezing (-80˚C) and anoxia. Anoxic environment was generated in a desiccation chamber at 60%RH by flushing the Nitrogen gas into the chamber. The concentration of oxygen inside the chamber was monitored. After each timepoint they were rehydrated with 500 µl of water for 2–3 hours. Rehydrated worms were transferred to NGM

agar plates with *E. Coli NA22* as food. Survivors were counted after overnight incubation at 15˚C. Each experiment was performed on two different days with at least two technical replicates.

## Trehalose quantification from nematode lysates

Trehalose measurements were performed as described in previous reports [27].

## Radiolabeling, metabolite extraction and 2D-TLC

The above-mentioned procedures were performed according to previous reports [27,28].

## Identification of trehalose-6-phosphate from TLC plates

Normalized aqueous fractions from the non-preconditioned and preconditioned samples were separated by high performance thin layer chromatography (HPTLC), using 1-propanol-methanol-ammonia (32%)-water (28:8:7:7 v/v/v/v) as first, dried for 15 min and 1-butanol-acetone-glacial acetic acid–water (35:35:7:23 v/v/v/v) second dimension respectively. Using the trehalose as a standard on both dimensions of the TLC, the regions of interest were scrapped out from the TLCs. The scraped-out silica was extracted with 10 ml of 50% methanol twice. The fractions were combined, dried under vacuum and dissolved in 100 μl of MS mix solution containing 4:2:1 (Isopropanol:Methanol:Chloroform) with 7.5 mM ammonium formate. Mass spectrometric analysis was performed on a Q Exactive instrument (Thermo Fischer Scientific, Bremen, DE) equipped with a robotic nanoflow ion source TriVersa NanoMate (Advion BioSciences, Ithaca, USA) using nanoelectrospray chips with a diameter of 4.1 μm. The ion source was controlled by the Chipsost 8.3.1 sostware (Advion BioSciences). Ionization voltage was + 0.96 kV in negative mode; backpressure was set at 1.25 psi. The temperature of the ion transfer capillary was 200˚C; S-lens RF level was set to 50%. FT MS spectra were acquired within the range of m/z 50–750 at the mass resolution of R m/z 200 = 140000; automated gain control (AGC) of $3\times10^6$ and with the maximal injection time of 3000 ms. FT MS/MS spectra were acquired within the range of m/z 50–750 at the mass resolution of R m/z 200 = 140000; automated gain control (AGC) of $3\times10^4$ and with a maximal injection time of 30 s.

## Triacylglycerols measurement from *P. kolymaensis* lysates

Non-preconditioned and preconditioned pellets were lysed in 200 μl of isopropanol with 0.5 mm Zircornium beads twice for 15 min. The lysates were centrifuged at 1300 g for 5 min. The supernatant was carefully collected without any debris, 20 μl of the lysate was used for protein estimation. Normalization was performed according to soluble protein levels, supernatant volumes corresponding to 50–100 μg of proteins were dried in the desiccator. 700 μl of IS ((10:3 (Methyl tert-butyl ether: ethanol)) mix (warmed to room temperature) was added to dried samples and left on the shaker for 1 hour. The samples were centrifuged at 1400 rpm and 4˚C. 140 μl of water was added and left on the shaker for 15 min. These samples were centrifuged at 13400 rpm for 15 min. The upper organic fraction was collected and transferred to 1.5 ml glass vial and left for drying in the desiccator. The dried samples were reconstituted in a volume of 300 μl of 4:2:1 (Isopropanol:Methanol:Chloroform). Volume corresponding to 1 μg was used for injection.

LC-MS/MS analysis was performed on a high-performance liquid chromatography system (Agilent 1200 HPLC) coupled to a Xevo G2-S QTof (Waters). The samples were resolved on a reverse phase C18 column (Cortecs C18 2.7um from Waters) with 50:50:0.1:1% (Water:Methanol:Formicacid:1MAmmoniumformate) and 25:85:0.1:1% (Acetonitrile:Isopropanol:Formic

acid:1M Ammonium formate) as mobile phase. The following gradient program was used: Eluent B from 0% to 100% within 12 min; 100% from 12 min to 17min; 0% from 17 min to 25 min. The flow rate was set at 0.3 ml/min. The samples were normalised according to the total protein concentration and the worm numbers. TAG 50:00:00 was used as internal standard. Skyline-software (https://skyline.ms/project/home/software/Skyline/begin.view) was used to analyse the raw data. TAGs were extracted from Lipidmaps (https://www.lipidmaps.org/) database.

## Supporting information

**S1 Fig. Calibration of a radiocarbon ($^{14}$C) date.** Radiocarbon date (44,315±405 BP) and calibrated age (45,839–47,769 cal BP) of plant material collected from buried borrow P-1320. Radiocarbon ages were converted to calendar age equivalents with the OxCal V.4.4 program using the IntCal20 calibration curve. Pink-shaded area—radiocarbon date with standard deviation; grey-shaded area—radiocarbon date projection on the calibration curve with 95.4% probability.
(PDF)

**S2 Fig. Morphology of *P. kolymaensis* female.** Graphic presentations of holotype (A, B) and SEM pictures (C-I): a) anterior body, B) female reproductive branch, C–E) anterior end of three different female specimens, F) anterior part of the lateral ridge, G) vulva, H) ventral excretory/secretory pore, I) posterior body with anus and lateral ridge. Scale bars: a,b—50 μm, c,i—3 μm, d—2 μm, e –1 μm, f,h—5 μm, g—10 μm,.
(PDF)

**S3 Fig. Phylogenies inferred for genes sets using both concatenation and coalescence approaches.** A) The ploidy level was analysed with *Smudgeplot* v0.2.1[73]. KMC version 3.1.0[74] was used to count the 21-mers in the PacBio CCS reads. Then we ran *smudgeplot.py* to determine the lower and upper coverage cut-offs. These were determined to be 14 and 380. 21-mers with a coverage between 14 and 380 were filtered with kmc_tools. Then, we computed the k-mer pairs from the filtered 21-mers by running *smudgeplot.py hetkmers*. Finally, the produced a smudgeplot shows an estimated ploidy of 3. B) Species tree inferred for 12,295 gene trees using coalescence approach. The species tree implemented using the coalescence approach with orthogroup gene tree is displayed. Novel species in this study are displayed in red. All nodes have a posterior probability of 1. C) Species tree inferred for 102 taxa. The maximum likelihood tree inferred using a concatenated supermatrix of 60 genes is displayed. Bootstrap values are only displayed for nodes with less than 100% support. The platyhelminth *Macrostum lignano* serves as an outgroup for rooting. The ancestral *Panagrolaimus*, sister to all others within the *Panagrolaimus* genus, is highlighted in red.
(PDF)

**S4 Fig. Combination of cryptobiotic states enhances survival of *C. elegans* dauer larvae.** A) Desiccated dauer larvae manifest enhanced survival rate to heat stress (34˚C). Error bars indicate standard deviation of two independent experiments with two technical replicates. B) Desiccated dauer larvae display enhances survival rate to anoxia. Error bars indicate standard deviation of two independent experiments with two technical replicates. Statistical comparison was performed by paired two tailed t-test. *p<0.05. C) *P. kolymaensis* possesses gene orthologs to most genes implicated in dauer formation and metabolism in *C. elegans*. Black filled circles: Ortholog presence suggested by orthogroup clustering, phylogenetic analysis, and domain architecture. White filled circles: No ortholog found via current analysis (in all cases these *C. elegans* genes did not cluster with any *Panagrolaimus* genes in the orthogroup clustering). Label: *C. elegans* enzyme names and orthogroup that contains that gene according to our

orthogroup clustering.
(PDF)

**S5 Fig.** *P. kolymaensis* **reduces triacylglycerols (TAGs) levels and accumulates trehalose-6-phosphate upon preconditioning at 98%RH.** A) 1D-Thin layer chromatography of acetate labelled organic fractions of non-preconditioned (1) and preconditioned (2) *P. kolymaensis.* B) Mass spectrometric quantification of TAG levels of non-preconditioned (1) and preconditioned (2) *P. kolymaensis.* Error bars indicate standard deviation of two biological replicates with two technical replicates. Statistical analysis was performed using unpaired t-test with Welch correction **p< 0.001. C-D) non-preconditioned and preconditioned mass spectrum of an empty region, e-f) spot 1 (trehalose), G-H) spot 7 (trehalose-6-phosphate) scraped out and extracted from the 2D-TLC.
(PDF)

**S6 Fig. Exploring a possible allopolyploid origin for extra proteins.** Gene-tree reconciliation was used to determine whether extra sets of proteins across orthogroups originate through auto- or allopolyploidy. Different copies of proteins (designated by '+' and '*') suggest an allopolyploid origin. Parthenogenetic species are highlighted in bold.
(PDF)

**S1 Text. Supplementary Information and methods.**
(PDF)

**S1 Data. Desiccated dauer larvae manifest enhanced survival rate to heat stress (34˚C).**
(XLSX)

**S2 Data. Desiccated dauer larvae display enhances survival rate to anoxia.**
(XLSX)

**S3 Data. Mass spectrometric quantification of TAG levels of non- preconditioned (1) and preconditioned (2)** *P. kolymaensis.*
(XLSX)

**S4 Data. Survival rate of** *P. kolymaensis* **nematodes to desiccation and freezing (-80˚C).**
(XLSX)

**S5 Data.** *P. kolymaensis* **nematodes and** *daf-2(e1370)* **dauer larvae upregulate trehalose levels upon preconditioning at 98%RH.**
(XLSX)

**S6 Data. Desiccated** *daf-2 (e1370)* **dauer larvae survive to freezing (-80˚C) for an extremely long period.**
(XLSX)

**S7 Data. Brood size of desiccated dauer larvae exposed to freezing remain like that non-desiccated dauer larvae.**
(XLSX)

**S1 Orthology analysis. Supplementary file orthology analysis.**
(PDF)

## Acknowledgments

VG is thankful to Andrej Shevchenko, members of the Volkswagen grant and Kurzchalia lab for helpful discussions and the core facilities of MPI-CBG for assistance. We are grateful to Dr.

S. Gubin for field study and sampling, our colleagues in Soil Cryology Lab, Pushchino and North-East Scientific Station in Chersky, Republic of Sakha (Yakutia) for their help and cooperation. The authors thank Long Read Team of the DRESDEN-concept Genome Center, DFG NGS Competence Center, part of the Center for Molecular and Cellular Bioengineering (CMCB), Technische Universität Dresden and MPI-CBG. We thank https://www.copernicus.eu/en for the base map in Fig 1A. The authors are thankful to Richard Roy and Jens Bast for critically reading the manuscript. We thank Iain Pattern for suggestions on writing the manuscript.

## Author Contributions

**Conceptualization:** Vamshidhar R. Gade, Philipp H. Schiffer, Teymuras V. Kurzchalia.

**Data curation:** Vamshidhar R. Gade, Martin Pippel, Tarja T. Hoffmeyer, Lewis Stevens, Graham M. Hughes.

**Formal analysis:** Vamshidhar R. Gade, Martin Pippel, Tarja T. Hoffmeyer, Alexei V. Tchesunov, Lewis Stevens, Graham M. Hughes, Philipp H. Schiffer.

**Funding acquisition:** Anastasia Shatilovich, Eugene W. Myers.

**Investigation:** Anastasia Shatilovich, Vamshidhar R. Gade, Martin Pippel, Tarja T. Hoffmeyer, Alexei V. Tchesunov, Lewis Stevens, Graham M. Hughes, Michael Hiller.

**Methodology:** Anastasia Shatilovich, Vamshidhar R. Gade, Martin Pippel, Tarja T. Hoffmeyer, Alexei V. Tchesunov, Lewis Stevens, Sylke Winkler, Graham M. Hughes, Sofia Traikov, Michael Hiller.

**Project administration:** Vamshidhar R. Gade, Philipp H. Schiffer.

**Resources:** Martin Pippel.

**Software:** Martin Pippel, Tarja T. Hoffmeyer, Graham M. Hughes.

**Supervision:** Vamshidhar R. Gade, Elizaveta Rivkina, Philipp H. Schiffer, Eugene W. Myers, Teymuras V. Kurzchalia.

**Validation:** Vamshidhar R. Gade.

**Visualization:** Vamshidhar R. Gade, Martin Pippel.

**Writing – original draft:** Anastasia Shatilovich, Vamshidhar R. Gade, Martin Pippel, Tarja T. Hoffmeyer, Graham M. Hughes, Philipp H. Schiffer, Teymuras V. Kurzchalia.

**Writing – review & editing:** Anastasia Shatilovich, Vamshidhar R. Gade, Martin Pippel, Tarja T. Hoffmeyer, Lewis Stevens, Sylke Winkler, Graham M. Hughes, Philipp H. Schiffer, Teymuras V. Kurzchalia.

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
