## [Decision Letter · Decision Letter 0]

6 Apr 2023

Dear Dr Kurzchalia,

Thank you very much for submitting your Research Article entitled 'A novel nematode species from the Siberian permafrost shares adaptive mechanisms for cryptobiotic survival with C. elegans dauer larva' to PLOS Genetics.

The manuscript was fully evaluated at the editorial level and by one of the original independent peer reviewers who also saw it at PCI Zoology. The reviewer is essentially satisfied with the revised manuscript, but has a couple of minor textual amendments - please pay particular attention to the issue of the Latin binomial since this will likely have long standing consequences.  When you have addressed these issues I will be able render a final editorial decision without further external evaluation.

We therefore ask you to modify the manuscript according to the review recommendations. Your revisions should address the specific points made by each reviewer.

Yours sincerely,

Gregory Copenhaver

Editor-in-Chief

PLOS Genetics

Reviewer's Responses to Questions

**Comments to the Authors:**

Reviewer #1: The authors now provide accession numbers to sequences, thank you.

A glaring error is failure to include the new species name in the manuscript. It is incorrect to name the species "n. sp." (as redundantly in line 163: "Box 1 Description of Panagrolaimus n. sp sp. nov." (also line 165) or line 189: Panagrolaimus sp. nov. Please also note that the "n. sp." should not be italicized. The authors already provide a species name near the bottom of box 1, "P. kolymaensis". Please use this name throughout the paper.

It is not correct to refer to a trait as "monophyletic". That is a term applied to taxa. What the authors mean is "apomorphic" (and instead of "polyphyletic", a trait would be "homoplastic").

**Have all data underlying the figures and results presented in the manuscript been provided?**

Reviewer #1: Yes

PLOS authors have the option to publish the peer review history of their article (what does this mean?). If published, this will include your full peer review and any attached files.

Reviewer #1: No

---

## [Editor Report · Decision Letter 1]

24 May 2023

Dear Dr Kurzchalia,

We are pleased to inform you that your manuscript entitled "A novel nematode species from the Siberian permafrost shares adaptive mechanisms for cryptobiotic survival with C. elegans dauer larva" has been editorially accepted for publication in PLOS Genetics. Congratulations!

Yours sincerely,

Gregory P. Copenhaver

Editor-in-Chief

PLOS Genetics

Gregory S. Barsh

Editor-in-Chief

PLOS Genetics

Comments from the reviewers (if applicable):

**Data Deposition**

http://datadryad.org/submit?journalID=pgenetics&manu=PGENETICS-D-23-00182R1

**Press Queries**

---

## [Editor Report · Acceptance letter]

4 Jul 2023

PGENETICS-D-23-00182R1 

A novel nematode species from the Siberian permafrost shares adaptive mechanisms for cryptobiotic survival with C. elegans dauer larva 

Dear Dr Kurzchalia, 

We are pleased to inform you that your manuscript entitled "A novel nematode species from the Siberian permafrost shares adaptive mechanisms for cryptobiotic survival with C. elegans dauer larva" has been formally accepted for publication in PLOS Genetics! Your manuscript is now with our production department and you will be notified of the publication date in due course.

With kind regards,

Anita Estes

PLOS Genetics

On behalf of:
